# Correlating Ultrastructural Changes in the Invasion Area of Colorectal Cancer with CT and MRI Imaging

**DOI:** 10.3390/ijms25189905

**Published:** 2024-09-13

**Authors:** Joanna Urbaniec-Stompór, Maciej Michalak, Janusz Godlewski

**Affiliations:** 1Department of Diagnostic Imaging, Clinical Hospital of the Ministry of Internal Affairs and Administration with the Warmia-Mazury Oncology Centre, 10228 Olsztyn, Poland; 2Department of Oncology, Faculty of Medical Sciences, University of Warmia and Mazury, 10228 Olsztyn, Poland; 3Department of Human Histology and Embryology, Faculty of Medical Sciences, University of Warmia and Mazury, 10082 Olsztyn, Poland; 4Clinical Surgical Oncology Department, Clinical Hospital of the Ministry of Internal Affairs and Administration with the Warmia-Mazury Oncology Centre, 10228 Olsztyn, Poland

**Keywords:** colorectal cancer, tumour microenvironment, cancer-associated fibroblasts, front of cancer invasion, desmoplastic reaction, computer tomography, magnetic resonance, cancer evaluation

## Abstract

The cancer invasion of the large intestine, a destructive process that begins within the mucous membrane, causes cancer cells to gradually erode specific layers of the intestinal wall. The normal tissues of the intestine are progressively replaced by a tumour mass, leading to the impairment of the large intestine’s proper morphology and function. At the ultrastructural level, the disintegration of the extracellular matrix (ECM) by cancer cells triggers the activation of inflammatory cells (macrophages) and connective tissue cells (myofibroblasts) in this area. This accumulation and the functional interactions between these cells form the tumour microenvironment (TM). The constant modulation of cancer cells and cancer-associated fibroblasts (CAFs) creates a specific milieu akin to non-healing wounds, which induces colon cancer cell proliferation and promotes their survival. This review focuses on the processes occurring at the “front of cancer invasion”, with a particular focus on the role of the desmoplastic reaction in neoplasm development. It then correlates the findings from the microscopic observation of the cancer’s ultrastructure with the potential of modern radiological imaging, such as computer tomography (CT) and magnetic resonance imaging (MRI), which visualizes the tumour, its boundaries, and the tissue reactions in the large intestine.

## 1. Introduction—General Features of Cancer Invasion

Colorectal cancer (CRC) is a result of mutations of protooncogene, suppressor, and mutator genes within epithelial cells of the intestinal mucous membrane. The reprogramming of genetic information causes continuing morphological and physiological transformation of these cells. Initially, colonic epithelial cells have their natural phenotype, which is characteristic of the healthy columnar epithelium. The uncontrolled proliferation of these cells and lack of apoptosis may result in specific structural changes within the intestinal mucosa, generally named polyps (tubular or villous adenomas), although neoplasm development without such transfiguration within the mucous membrane is also possible. During this dysplastic stage, the transformation of epithelial cells is referred to as the epithelial-to-mesenchymal transition (EMT) phenomenon, and as a result, cells gradually acquire new characteristics: oval or variable shape and size, lack of cellular polarization, as well as a loose and irregular arrangement within the mucous layer. One of the main features acquired by the transformed cells in this phase is the production of digestive enzymes which degrade the ECM components [1]. Due to this, neoplasm cells achieve the ability to cross through the basement membrane (BM) and next, they can destroy lamina propria and subsequently invade the submucosa layer. The lamina propria involvement is defined as the conversion from the stage of cancer in situ into an invasive cancer form.

As the process of CRC invasion in patients’ population varies, the aim of modern personalised medicine is to identify the patients with a more aggressive course of the disease to administer appropriately intensive treatment. The aim of this review is to present the process of CRC promotion and progression at the ultrastructural level and its relation to modern diagnostic imaging (CT, MRI) routinely performed at the initial stage of the diagnosis. The presentation of histological features and its reference to the macroscopic image may be helpful at the initial diagnosis with the selection of appropriate treatment.

## 2. CRC Extracellular Matrix Disintegration Due to Metalloproteinase Activity

The gradual destruction of the intestinal wall layers is caused by the invasion of cancer cells, which have the ability to produce and secrete proteases. Initially, these proteolytic enzymes enable cancer cells to migrate from the epithelium through the basement membrane and lamina propria, a structural component of the mucosa membrane. These two elements are built up from connective tissue—fibres and ground substance. The basement membrane (sheetlike network) is composed mostly of type IV collagen, and lamina propria is composed of a loose arrangement of type I and type III collagen fibres [2]. Matrix metalloproteinases (MMPs, 23 enzymes in human tissue) degrade different types of collagen structures. Types I, III, and IV collagen are degraded mainly by MMP-1 (which belongs to the gelatinase type), MMP-2, and MMP-9 (collagenase type) proteolytic activity [3]. Cancer cells produce and release MMP enzymes into adjacent connective tissue to degrade the ECM and improve cell migration. There are different modes of this enzyme secretion. For example, MMP-1 is located in extracellular vesicles shed into the tissue stroma, and MMP-14 is localised on the cell surface, mostly in the invadopodia area [1,4]. It was confirmed that colorectal cancer tissues (cancer cells and stroma cells) show elevated immunoreactivity of MMP-1, and the intensity of its expression is correlated with the depth of cancer invasion into intestinal structures [5]. Numerous experimental studies based on gelatine zymography and immunohistochemistry methods have demonstrated higher expression of mainly MMP-2 (inactive pro-MMP-2 and active MMP-2 forms) and MMP-9 (pro-MMP-9 and MMP-9) enzymes within colorectal cancer mass and dysplastic adenomas compared to the corresponding normal mucosa of the large intestine [6,7,8,9]. MMP-9 cleaves type IV collagen, ref. [3] so the increased expression of this enzyme is observed in the area of cancer invasion through the basement membrane, where the reduced presence of type IV collagen is observed simultaneously [7]. The fact that MMP-9 immunoexpression is observed in the infiltrative tumour border configuration supports the thesis that such proteolytic activity promotes the aggressive potential of cancer cells [10].

Compilated data from numerous studies presented in a review by Pezeshkian et al. show that the process of neoplasm transformation from benign adenoma into malignant adenocarcinoma involves the activation of a higher number of MMPs. In the phase of adenoma development, increased expression of MMP-1, MMP-3, MMP-7, and MMP-9 was observed, and subsequently, at the stage of colorectal cancer invasion, other MMPs, including MMP-2, MMP-8, MMP-10, MMP-12, MMP-13, MMP14, and MMP-21 are activated and overexpressed [11]. Several studies have demonstrated a strong correlation between the expression of MMP-1, MMP-2, MMP-7, MMP-9, MMP-10, and MMP-12 enzymes, tumour size, cancer progression, lymphatic dissemination, and bloodstream distant metastasis (table in Section 5) [5]. Among others, the expression of MMP-2 and MMP-9 proteins in CRC tissue as potential biomarker roles is underlined due to their correlation with important clinical data: cancer grade, higher risk of dissemination or disease recurrence, and shorter patient survival [12,13,14].

Disintegration or decomposition of the intestinal basement membrane, lamina propria, and submucosal membrane due to cancer cells’ proteolytic activity does not only mean the degradation of collagen fibres which are present in this area, as ECM is a highly composed environment where the ground substance, numerous proteoglycans, glycosaminoglycans (heparan sulphate), and multi-adhesive glycoproteins (laminin, fibronectin) are present. Fragmentation of the epithelial ground substance and its components by MMPs release growth factor deposits which are constantly present in the ECM area: transforming growth factor-β (TGF-β), fibroblast growth factor (FGF), and insulin-like growth factor (IGF) [15,16,17].

TGF-β latent complex is cleaved into mature cytokine form via protease enzyme activity or by integrin-dependent activation, which means interaction with these transmembrane adhesion molecules [18,19].

The decomposition of collagen fibres is a well-known biological fact that occurs during the wound-healing process. Early scar tissue remodelling takes place through fibroblast activity, which secretes MMPs. In normal conditions, proteinases MMP-2 and MMP-9 reduce tissue fibrosis and stimulate re-epithelialisation processes at the leading edge of wounds [20,21]. During wound healing, MMP-dependent ECM proteolysis is controlled by the activity of endogenous tissue inhibitors of metalloproteinases (TIMPs), which are in relative balance [22]. However, in the colorectal cancer invasion area, the lack of TIMP-1 (an MMP-9-specific inhibitor) expression may indicate that MMP-9 enzyme activity causes proteolytic degradation of the ECM components without any inhibition or control of connective tissue remodelling [23]. The disintegration processes of ECM within the large intestine wall additionally comprise the activation and migration of connective-tissue-resident cells. These enzymes could originate from all the cells that cooperatively constitute the TM: cancer cells, tumour-associated macrophages, and CAFs [24].

## 3. CRC Tumour Microenvironment

In the area of cancer invasion within the intestinal wall, the varied population of residents (which belong to the connective tissue) and transient cells (mostly of immune origin) are present. Cancer cells interact mainly with mesenchymal stem cells, myofibroblasts, and macrophages via different modulation modes: cell-to-cell and/or cell-to-ECM reaction, receptor signalling due to soluble ligands (growth factors), or even changes to the tissue pressure. The accretion of macrophages and fibroblasts in this area creates a specific microenvironment similar to a non-healing chronic wound.

### 3.1. TAMs—Tumour-Associated Macrophages

Under normal conditions, macrophages (which belong to the mononuclear phagocytic system) are recognised as resident or transient cells. Activated macrophages, e.g., during infection, are divided into two types which play their role in two different stages of the inflammatory process: pro-inflammatory M1 macrophages, which initiate immune response, and M2 macrophages (mainly M2C subgroup), which reduce the inflammatory process, promote tissue remodelling and repair, and generally stimulate wound healing [25].

The tumour microenvironment (TM), probably due to low oxygen pressure and the presence of tissue debris, attracts monocytes via chemokines CCL2, CCL5, vascular endothelial growth factor (VEGF), macrophage colony-stimulating factor (M-CSF), and TGFβ-activity, and generates changes in the population of these cells. Macrophages present in the TM area are named TAMs, with a significant predominance of M2 macrophages, which are polarised by interleukin 2 (IL-2), IL-4, IL-13, IL-10, and TGF-β stimulation [26,27,28]. Studies concerned with TAM presence in a TM area, especially at the front of invasion, demonstrate that the high number of M2 macrophages or changed M2/M1 ratio (M2 macrophage predominance) creates a more invasive potential of neoplasm and has correlated with the presence of metastases. This fact reveals that M2 macrophages in the TM area could be a predictive factor concerning the worst course of illness [29]. Activated M2 macrophages produce numerous chemokines and growth factors that induce numerous tumour-promoting processes. For example, TGF-β, FGF2, EGF, and interleukin 6 (IL-6) stimulate cancer cell proliferation; VEGF and platelet-derived growth factor (PDGF) promote angiogenesis; MMP-2 and MMP-9 degrade extracellular stroma; and CCL22 recruits Tregs lymphocytes to cause tumour immunosuppression [26].

Generally, if M2 macrophages are activated in TM, they become involved in cancer cell invasion promotion, despite their physiological pro-healing role. Considering the numerous interactions of M2 macrophages, their impact on fibroblast activity seems to be crucial in neoplasm development.

### 3.2. CAFs—Cancer-Associated Fibroblasts

Normally, quiescent fibroblasts are present within the intestinal connective tissue, mainly in the lamina propria, and they produce and maintain ECM components. In the course of chronic or acute intestinal inflammation with moderate tissue damage, the macrophage/fibroblast feedback is critical. During the regeneration and repair processes, activated fibroblasts transform into myofibroblasts by numerous signalling molecules generated by M2 macrophages (TGF-β, PDGF, IGFI, and IL-4) and fibronectin, then migrate and intensively produce ECM components, mainly collagen fibres, which finally cause intestinal wound healing [30].

The key axis of the regeneration process (stroma formation, remodelling, and angiogenesis) has been identified as the main feature of cancer invasion. Moreover, the above-described macrophage–fibroblast interactions occur during cancer invasions. In the TM, activated fibroblasts, myofibroblasts, and mesenchymal stem cells comprise a cellular group defined as cancer-associated fibroblasts (CAFs) [31]. These connective tissue cells are activated and transformed into CAFs by mechanical tension, cancer cell interaction, and M2 macrophage secretion, mainly due to TGF-β signalling but also via CXCL12, PDGF, IL-4, IL-6, and IGF-1 activity [32,33,34].

Generally, CAFs have a significantly higher cell potential, leading to increased proliferation, more intensive cell migration, and the secretion of growth factors, chemokines, and cytokines. Summarizing the research to date allows this cell population to be defined as a group that stimulates numerous mechanisms related to the promotion of cancer development [35]. CAFs produce and secrete numerous growth factors (TGF-β, FGF, HGF, PDGF, EGF, and IGF 1/2) that also stimulate the proliferation of colon cancer cells and significantly inhibit apoptosis (Figure 1) [36,37].

TGF-β is a good example of the complexity of the interaction between cells which are present in the TM. In normal conditions, TGF-β signalling involves type I and type II serine/threonine kinase receptors, which, after dimerization, become phosphorylate SMAD 2/3 and SMAD 4 proteins which subsequently translocate to the nucleus as trimers to bind with the target DNA.

This transcription factor (SMAD) negatively controls and inhibits cell proliferation and growth via such canonical signalling pathways. During neoplasm development, TGF-β signalling is disrupted and non-canonical, SMAD-independent pathways are promoted: mitogen-activated protein kinase (MAPK), phosphoinositide 3-kinase (PI3Κ)/Akt, and Rho/Rho-associated protein kinase (ROCK). These kinds of receptors and these pathways are crucial factors in neoplasm development and have a significant impact on cancer cell proliferation and survival [38]. Furthermore, it should be noted that TGF-β is secreted by the cells of TM (cancer cells, TAMs, and CAFs) with autocrine and paracrine signalling, and finally, such multifactorial positive feedback promotes invasive CRC phenotypes at the initial and advanced stages of illness [39,40].

## 4. Neoplasm Tumour Organisation and Stiffness

The processes briefly described above are concerned with interactions at the molecular and cellular levels. Nevertheless, equally important changes are related to the ECM components and apply to the organ’s ultrastructure, which subsequently alters the large intestine morphology and function. Aggressive cancer cells, via protease activity, digest and destroy the proper structure of the large intestine, gradually replacing normal tissue structure with tumour mass. Since increased CAF activity predominantly causes higher production and secretion of collagen molecules, enhanced deposition of ECM components, especially collagen fibres and their slower turnover, is observed within the neoplastic tumour [41]. The spatial organization of collagen fibres differs significantly from the arrangement of the proper fibres in normal tissues [42]. In addition to differences in the quantity and alignment of collagen, alterations in the molecular characteristics and cross-linking of fibres are found, as well as an increase in the amount of glycosaminoglycans (GAGs), heparan sulphate, and chondroitin sulphate within TM [43].

All of these changes within the ECM can promote colon cancer cell aggressiveness. Experimental studies demonstrate that tumour cells grow significantly faster in TM matrix conditions than in normal ECM [44], and altered stiffness and matrix organisation due to different types of type I collagen and their concentration could also activate colorectal cancer cell migration [45].

Cancer cells interact with stroma components through the integrins (adhesion glycoproteins). These transmembrane glycoproteins are mechanotransductors which transmit ECM mechanical signals into the cytoplasm. They activate actin filament (cytoskeleton), focal adhesion kinase (FAK), and SRC family kinases (SFKs) signalling, and subsequently stop contact inhibition phenomena (YAP/TEAD) signalling. Integrin, via a synergistic mechanism, interacts with receptor tyrosine kinases (RTKs) which initiate crucial pro-mitotic Ras-ERK, PI3K/AKT pathways as well as inhibit apoptosis. In summary, the impact of these multimodal signals results in cancer cell migration, proliferation, and survival [46,47].

Moreover, an important feature of cancer cell growth is the lack of contact inhibition [48], so they consistently proliferate, forming a cancerous mass of tightly packed cells. These cells are located between dense alignments of collagen fibres, and this spatial network of cancerous tissues generates lower elasticity and higher density of neoplasm tumours [49,50].

This tissue stiffness is so characteristic of the neoplasm structure that the terms “tumour” and “malignancy” are synonymous. In clinical practice, it is usually observed that growing cancerous tumours within the intestinal wall (plus mass protruding into the lumen) create a rigid ring-shaped narrowing of the left side of the large intestine, which causes a mechanical obstacle to the faeces passage. The stiff circular narrowing of the intestinal lumen affects a change in large intestine movements, subsequently leading to sub-obstruction and next-to-complete obstruction of the large intestine.

## 5. Cancer Invasion and Desmoplastic Reaction

As described above, the proteinase effect and the predominance of MMP enzymes in relation to their down-regulated inhibitors (TIMPs) cause significant ECM disintegration, which cancer cells use to create a “front of invasion.” Upon the pathomorphological observations, two different types of cancer cell invasion into the intestinal wall were distinguished and defined. The first form is an “infiltrative pattern”, which means the pervasion of cancerous cords into normal tissues (Figure 2).

Usually, singular cells and small aggregation of cancer cells, tumour buds (less than five cancer cells), and/or poorly differentiated clusters (equal or more than five cells) are noticeable in the peripheral area at the front of these irregular neoplasm formations. The most important feature is the lack of a transparent boundary line between cancerous and normal tissues in the low magnification (40×) of microscopic observation [51,52,53].

The next form is an “expansive pattern”, with a well-demarcated tumour mass which pushes to the side of the normal structure of the large intestine (Figure 2). The boundary between the tumour mass and normal tissue is formed by collagen fibre aggregation [51,54].

These two different modes of cancer invasion result in diverse molecular backgrounds of cancer cell potential and aggressiveness, which consequently lead to various courses of this illness. The infiltrative tumour border configuration is mostly associated with blood vessel invasion, lymph node metastases, and distant spread into internal organs. This pathomorphological observation is an independent factor that is correlated with poor patient prognosis, shorter overall survival, and reduced disease-free survival [55]. It is significant that the presence/absence of tumour budding has now been included in the pathomorphological postoperative evaluation [56].

The myofibroblast and CAF activity in reaction to the cancer cell invasions creates an abundance of connective tissue fibres in the vicinity of tumour infiltration. These borderline areas have been described by Ueno et al. (2021) as the fibrotic cancer stroma with its outermost leading edge, which is defined as the desmoplastic front. Their pathomorphological observation allows the differential morphology of this desmoplastic front to be evaluated. The presence of myxoid stroma (amorphous mucinous substance), keloid-like collagen (thick collagen bundle eosinophilic hyalinization), and fine, multilayer collagen fibres were described as immature, intermediate, and mature desmoplastic reactions, respectively [57]. There are significant differences in the occurrence of particular types of tissue entanglement. Kobayashi (2023), in the microscopic study of a large group of 443 patients, demonstrated that immature, intermediate, or mature desmoplastic reactions are observed in 64%, 21%, and 15% of neoplasm, respectively [58]. Distinguishing these particular types of the desmoplastic front is important during the evaluation of postoperative material because an immature type of desmoplastic reaction is correlated with a colorectal cancer patient’s poor prognosis (higher pT stage, lymphatic/blood vessels invasion, and regional lymph node dissemination) (Table 1) [58].

As depicted above, the microenvironment of cancer tumours is very diverse from the histological point of view; these tissues have different organisation, alignment, stiffness, and density, which significantly determines a more invasive cancer phenotype. The inquiry is whether this differentiated tissue ultrastructure, arrangement, and infiltration can be detected by currently used radiological imaging examinations.

## 6. Modalities in Modern Radiology of Colorectal Cancer

The gold standard for an initial diagnosis of large intestine neoplasm is colonoscopy, which allows several crucial features to be evaluated: the exact localisation and distance from the anal sphincter, the size of the tumour, and, additionally, the preservation of intestinal lumen passage. Moreover, it enables a biopsy to collect material for the histopathological confirmation of cancer presence [59]. Next, the diagnosis should be followed by the evaluation of the local and regional colorectal cancer invasion and potential cancer-distant metastases, which is possible using radiological imaging methods: computer tomography (CT) and magnetic resonance imaging (MRI) [60].

In clinical practice, CRC advancement is defined according to the three main features of the neoplasm process: the size of the tumour (T), lymph node (N), and metastases (M), which are rated by a unified coding system. The TNM reporting system is provided by the Union for International Cancer Control (UICC) and the American Joint Committee on Cancer (AJCC) [61]. The combination of CT and MRI is commonly used for preoperative assessment of CRC, and it has been assumed that cancer staging evaluated with diagnostic imaging should be matched with the letter i (image) before the abbreviation TNM [62].

## 7. Computer Tomography (CT) in CRC

CT is an imaging modality which uses X-ray beams that are attenuated while crossing the human body structures. The differences in X-ray beam absorption are registered by detectors, and collected data are converted into images, which illustrate tissue density measured in Hounsfield Units (HUs). The Hounsfield scale is a linear, quantitative measurement system for describing the human body tissues and fluid radiodensity. These are differentiated in a range from −1000 HUs (air) through 0 HUs (water) and +50 HUs (muscle) up to compact bone (+1900 HUs) density in CT scans [63].

The chest, abdomen, and pelvis contrast-enhanced CT is a standard modality used for the following CRC preoperation characteristics: T- (localisation and size of tumour), and N- and M-staging evaluation. It is also useful to assess potential local, regional (peri-intestinal invasion, tumour bulging, and fat stranding), and distal complications (tumour lateroconal fascia invasion, mesenteric stranding, abscesses, fistulas, and vascular occlusion) as well as other organ abnormalities [64]. Tumour size characteristics such as tumour length, maximum diameter, and volume have been proven to be correlated with pathologic lymph node metastasis and higher N-stage. [65] Preoperative T-stage is also the 5-year overall survival predictor of the patients, with survival rates for T1, T2, T3, and T4 being 96%, 89%, 75%, and 79%, respectively [66].

Local T-staging of colon cancer invasion using both CT and MRI is challenging. The colon has an anatomical three-dimensional orientation within the abdominal cavity, and physiological peristalsis causes constant movement of the large intestine. Furthermore, CT imaging has limited capability to demonstrate the particular layers of the intestinal wall because of reduced soft-tissue contrast resolution. Consequently, this causes difficulty in differentiating tumour cells invasion within the submucosa and muscularis layer (T1 vs. T2 tumour) and adds a limitation to evaluating the cancer protrusion beyond muscularis propria into the mesorectum area (T3, T4) [60,67].

Another difficulty concerning CT imaging is the different thicknesses of the intestinal wall, which may be related to individual normal variants, inflammatory processes, or obstruction due to stenosis caused by the tumour. The eventual intestinal thickness needs to be analysed in relation to the degree of thickening, symmetric versus asymmetric, focal/segmental or diffuse involvement, and presence of associated peri-intestinal abnormalities. Smaller tumours are visualised on contrast-enhanced CT scans as asymmetric or circumferential areas of intestinal wall thickening with homogeneous enhancement. Larger tumours frequently undergo central necrosis and demonstrate heterogeneous enhancement and an irregular bulky mass protruding into the intestinal lumen. Mucinous adenocarcinomas often show heterogeneous attenuation on contrast-enhanced CT scans because they contain poorly defined central areas of low attenuation related to intracellular tumour mucin deposition [68].

The CRC CT staging may be improved with widely available multiplanar reformats, allowing the generation of true axial images through the rectum [69].

A meta-analysis conducted by Nerad et al. demonstrated that contrast-enhanced CT (slice thicknesses <5 mm) is able to demonstrate tumour invasion through the intestinal wall (T1–T2 vs. T3–T4) with a sensitivity and specificity of 96% and 70%, respectively [70]. Another study which compared contrast-enhanced CT efficiency with postoperative pathomorphological microscopic evaluation revealed that T-staging was consistent with a high 80% overall accuracy. It has been emphasised that differences in CT exams were more frequently related to down-staging rather than over-staging. This study identified sources of these differences: ultra-small components are not visible at the contemporary level of CT in reference to the microscopy potential (minimal cancer invasion through the muscularis propria), and secondly, the mucinous neoplasm component (down-staging) and extensive immune reaction adjacent to the tumour (over-staging). The differentiation between an inflammatory reaction and cancer invasion, both with intensive cellular accumulation, is a well-known diagnostic problem of CT colon cancer imaging [71].

The accuracy of T3 and T4 diagnosis may also be increased with CT colonography (CTC, known as virtual colonoscopy), which is a contrast-enhanced CT study in which the colon is distended with low-pressure carbon dioxide supply via the rectum for more accurate visualisation. Scans are performed both in a supine and prone position and, later, 2D and 3D endoluminal reconstructions are made [72]. CTC shows high sensitivity and specificity for the detection and location of colon tumours, which are 96.63% and 71.79% [73], and high accuracy in staging with 98% and 84% sensitivity and specificity for T3-stage disease and 85% and 98% for T4-stage disease [74].

## 8. Magnetic Resonance Imaging (MRI) and CRC

MRI uses magnetic fields and radio waves to produce images. When the patient is placed in a strong magnetic field of an MRI scanner, the proton axes line up. This uniform alignment creates a magnetic vector oriented along the axis of the MRI scanner. When additional energy in the form of a radio wave is added to the magnetic field, the magnetic vector is deflected. Subsequently, when the radiofrequency (RF) source is switched off, the magnetic vector returns to its resting state (relax), and this generates a later detected signal. The intensity of the received signal is then plotted on a greyscale, and cross-sectional images are created. Different tissues (such as fat and water) have different relaxation times and can be identified separately [75].

MRI examination of the large intestine has substantial limitations. Firstly, the intraperitoneal parts of the large intestine (transverse and proximal sigmoid colon) have significant mobility and physiological peristalsis. Secondly, a long exposure time is needed to complete all of the scans from the abdomen and pelvis area, and all MRI sequences are very motion-sensitive. These conditions cause satisfactory quality large intestine imaging to be impossible to achieve, and MRI is not recommended in colon cancer assessment. On the contrary, the major part of the rectum has an extraperitoneal location and is fixed in the pelvic position, and the area of examination is significantly smaller, so MRI is useful and recommended as a standard procedure for rectum cancer imaging and its pretreatment staging [60].

MRI is an imaging method which may demonstrate with high precision gradual cancer invasion through the rectal muscularis propria, and enable detailed analysis and measurement of the tumour expansion. The clinical evaluation of rectal cancer based on radiological imaging examination is essential to differentiate patients with T1 and T2 tumours, who should be treated only by surgery, and patients with locally advanced T3 stage tumours who should obtain neoadjuvant radio-chemotherapy before large intestine resection [76,77].

The European Society of Gastrointestinal and Abdominal Radiology (ESGAR) specified several recommendations that T-stage MRI rectal cancer pretreatment evaluation should include: the tumour localisation and distance from its lower margin to the anorectal junction, size and length of tumour, presence of cancer deposits within the mesorectum, and an involvement of the mesorectal fascia (MRF) [78]. Next, in 2016, ESGAR experts made several changes to the recommendations based on their collective experience. They noted that extramural vascular invasion (EMVI), as well as circumferential growth of the tumour (e.g., from X to Y o’clock) within the rectal wall, should be mandatorily reported. Next, the slice thickness of exam ≤ 3 mm (in axial plane), and T-substages which describe depths of extramural invasion were added. Finally, the tumour position in relation to the anterior peritoneal reflection should also be evaluated [79].

Evaluation of particular layers of the rectal wall in MRI is difficult, as in the routine T2-weighted MRI (basic pulse sequence dependent on long echo time and long repetition time), since the rectum wall thickness is only 2–3 mm. A study performed by Bogveradze et al. demonstrated that MRI does not separate mucosa from submucosa, so only a single intermediate signal layer surrounded by the T2-hypointense layer that represents the muscularis propria is visible (two layers instead of a three-layered appearance). They also observed that mucosa and submucosa can be recognised as separate layers in the presence of submucosal oedema, only. In these cases, the submucosa could be visualised between the mucosa and muscularis propria as a middle high-signal layer [80]. Another study, performed by Taylor FG et al., showed that layers of the rectum can be identified. They described the mucosal layer as a fine low-signal-intensity line in T2-weighted images, with the thicker, high-signal submucosal layer seen just beneath. These layers are followed by the muscularis propria, which is depicted as two distinct layers; the inner circular layer; and the outer longitudinal layer. The outer muscle layer can be distinguished as an irregular grooved appearance with interruptions where vessels enter the rectal wall. In clinical practice, the limited visibility of the particular layers of the rectal wall is the main reason why MRI is generally unable to discern T1 from T2 tumours, which are often reported as T1–2 stage. MRI diagnosis of a T3-stage cancer invasion is based on the presence of a tumour signal extending into the perirectal fat with a broad-based bulging or nodular configuration in continuity with the intramural part of the cancer. Penetration into the muscular layers by small vessels and/or desmoplastic reaction are common pitfalls that can lead to over-staging T2 tumours as T3 tumours [81]. A desmoplastic reaction is depicted as spicules with low signal intensity, while T3 tumours have a broad-based or nodular appearance with intermediate signal intensity at T2-weighted imaging [62]. Because of the high resolution of MRI imaging in rectal localisation of cancer, the definition of the T3-stage of rectum tumour remains disputed (which is defined as the depth of tumour invasion through and out of muscularis propria), and three subclassifications are proposed. In the AJCC reporting system, the T3 stage has been subdivided into T3a, which is defined as an invasion into perirectal tissue ≤5 mm, and T3b, as an invasion >5 mm from the outer muscularis propria margin. The Union for International Cancer Control (UICC) propose a T3b stage interval from 5 mm up to 15 mm and adds the T3c stage, which is defined by the depth of cancer invasion into perirectal tissue more than 15 mm measured from the outer muscularis propria margin [67]. The most detailed estimation staging is proposed by the European Society for Medical Oncology (ESMO) and ESGAR, which provides four T3 substages based on depth of cancer invasion into perirectal tissue from the outer muscularis propria margin: T3a < 1 mm, T3b = 1–5 mm, T3c = 5–15 mm, and T3d > 15 mm (Figure 3) [67,82].

Cancer invasion into the rectal wall shows slightly higher signal intensity on T2-weighted MRI images to the relatively low signal intensity of the muscular layer and lower signal intensity of the mucosa/submucosa layer and mesorectal fat. These neoplasm tumours demonstrate variable enhancement after intravenous gadolinium-based contrast administration and may show restricted diffusion on diffusion-weighted imaging (DWI).

In MRI T2-weighted images, the perirectal fat appears as a high signal surrounding the low signal of the muscularis propria and contains signal void vessels. The distantly located mesorectal fascia is seen as a fine, low-signal layer enveloping the perirectal fat, and this layer defines the surgical excision plane in total mesorectal excision [83].

DWI is a functional non-invasive MRI technique based on the free Brownian motion of water molecules in different tissues. DWI detects and highlights the differences in the mobility of water molecules in the extracellular space of biological tissues. Hypercellular tissues such as tumours limit the diffusion of water molecules. The apparent diffusion coefficient (ADC) is derived from DWI and represents a calculated value that quantitatively reflects the restriction of diffusion [84]. A study performed by Yacheva et al. revealed that T3- and T4-stage tumours showed significantly lower ADC values and higher restriction of diffusion when compared to T1- and T2-stage tumours (*p* < 0.05) [85].

MRI enables the distinction of the mucinous subtype of neoplasm. Kim et al. differentiate this cancer based on an intrinsically high T2W signal of mucin presence with an accuracy > 95% [86]. It is important to identify these tumours because they have a worse course of illness and patient survival prognosis than non-mucinous cancers [62].

In conclusion, MRI is a superior diagnostic modality compared to CT for detecting T3 and T4 disease and EMVI, as it has better soft-tissue contrast than CT (Table 2). It provides scanning with high-resolution techniques and allows visualisation of the layer structure of the large intestine wall [87].

## 9. Radiomics and CRC

Radiomics corresponds to the digital extraction and analysis of numerous quantitative imaging features from conventional imaging modalities (CT or MRI) from the area of interest. The imaging information is analysed with specific clinical conditions and may be correlated with chosen data, such as tumour histology and cancer classification. It is able to provide more precise information than traditional imaging methods. It can be used not only for preoperative diagnosis, including TNM-staging, but also for prediction of therapeutic response and clinical outcome [89]. It is important to correctly delineate the ROI (region of interest) to be analysed. Segmentations must be reproducible and reliable. Automatic methods are preferable for precision as they are more precise compared to manual delineation [90].

Contemporary studies are exploring the applications of radiomics in the precise evaluation of tumour invasion and T-staging. The results of a study by Sun Y. et al. demonstrated that the T1–2 and T3–4 stages could be identified using radiomic features not only with an unsupervised method of clustering analysis but also with the LASSO (supervised machine-learning) method. T-staging of a rectal cancer using radiomics was more accurate than staging performed by two experienced radiologists using MRI. In addition, MRI staging for the T stage has lower sensitivity and specificity than radiomics. However, radiomics failed to demonstrate a significant performance in the evaluation of perineural invasion, histological grade, lymph–vascular invasion, and tumour deposits, which may be due to the limited sample sizes and imbalanced population distributions [91].

Another study conducted by Bo Deng et al. aimed to preoperatively differentiate T2- and T3-stage rectal cancer patients based on radiomic features of mesorectal fat in T2WI, DWI, and ADC sequences. They developed a theory that fat cells in the mesorectum activated by tumour cells secrete growth factors, adipokines, and ECM remodelling factors, which may manifest as changes in the expression profile of adipocytes around the tumour, which can lead to changes in MRI signals. The nomograms had good predictive value for preoperative T2 and T3 stages, with AUCs of 0.921 and 0.889 for the training and validation groups, respectively. It also indicated that ADC exhibits a strong correlation with the proliferation index KI 67, while true diffusion tends to correlate with cell count and KI 67 [76].

In the future, MRI applications could include blood oxygenation level dependent (BOLD) MRI and tumour oxygenation level dependent (TOLD) MRI, which could have a potential to identify, spatially map, and quantify tumour hypoxia, which occurs as an effect of a disordered angiogenesis [92]. After improving standardisation, validation, and reproducibility, and with prospective multicentric study trials, CT and MRI radiomics may become introduced to the clinical routine, to help staging the patients who require neoadjuvant chemotherapy [93].

## 10. Conclusions

Contemporary imaging methods (CT and MRI) are employed at the initial phase of patient assessment and demonstrate clinical stages of neoplasm development (locoregional vs. metastatic illness), as well as visualize size and cancer infiltration throughout the intestinal wall. These methods are crucial during clinical decision-making regarding cancer management and choosing the treatment approach: surgery, radiotherapy, and/or neoadjuvant chemotherapy. The preoperative staging of local cancer advancement (the T parameter of the TNM staging system) and type of cancer invasion into the large intestine wall allow for the determination of the neoplasm’s development and the prediction of the course of illness. This information is extremely important in the decision to implement adjuvant chemotherapy. The present review indicates that all diagnostic methods complement each other and have different roles and significance.

The limitations of CT and MR imaging methods are associated with difficulties in identifying the various layers of the intestinal wall and longitudinal cancer infiltration. The presented microscopic studies of the type of cancer invasion (expansive vs. infiltrative pattern) and the intensiveness of desmoplastic reaction should provide a reference point for further development of diagnostic imaging methods.

## Figures and Tables

**Figure 1 ijms-25-09905-f001:**
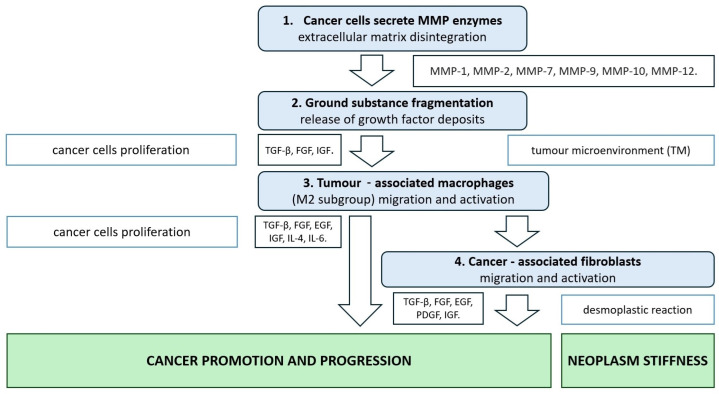
The role of enzymes and signalling molecules involved in promotion and progression of CRC.

**Figure 2 ijms-25-09905-f002:**
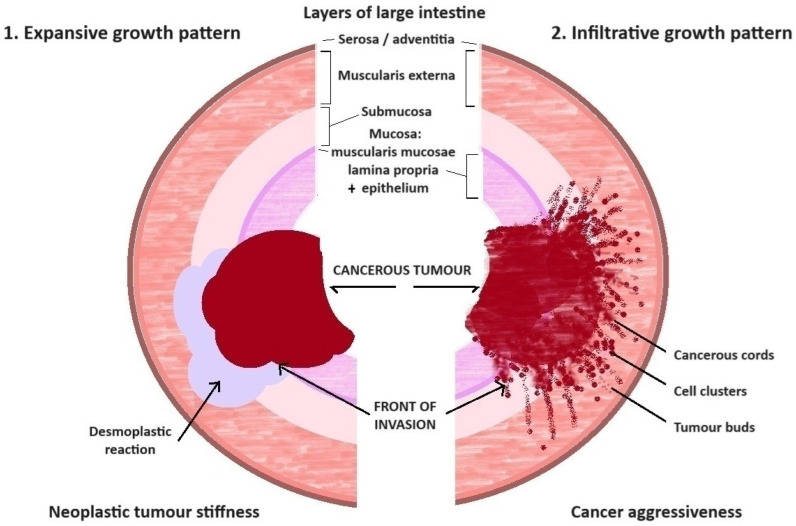
Comparison of two different growth patterns of large intestine cancer. The first one is expansive, with a well-demarcated tumour mass boundary, which pushes away the healthy tissue. The second one is infiltrative, without a demarcated mass boundary, with cords of cancer cells invading the healthy tissue.

**Figure 3 ijms-25-09905-f003:**
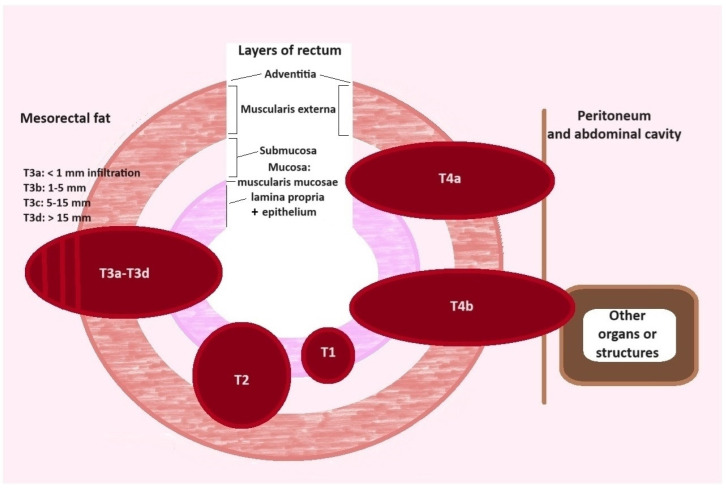
Illustration of different T-stages of CRC and their relation to layers of rectum.

**Table 1 ijms-25-09905-t001:** The potential molecular and cellular biomarkers and histological features observed in the pathomorphological assessment. Chosen features are related with colorectal cancer invasion, finally they are predictors of patients’ shorter survival.

Potential Biomarker	Pathomorphological Features Related with Colorectal Cancer Invasion
Elevated expression of metalloproteinase enzymes: MMP-1, MMP-2, MMP-7, MMP-9, MMP-10, and MMP-12	Correlation with tumour size, lymphatic dissemination, higher risk of metastases, and shorter survival [5,10,12,13,14].
M2 macrophages (mainly M2C subgroup) accumulation	Correlation with more invasive cancer potential and the presence of cancer-distant metastases [25,29].
The infiltrative tumour border arrangement and the tumour budding presence	Association with blood vessel invasion, lymph node metastases, and distant spread into internal organs. These features are correlated with shorter survival [55,56].
The immature type of desmoplastic reaction in the front of cancer invasion	Correlation with tumour size, lymphatic and blood vessels invasion, regional lymph node dissemination, and poor prognosis [58].

**Table 2 ijms-25-09905-t002:** Analysis of potential applications and limitations of chest, abdomen, and pelvis CT with intravenous contrast agent and abdomen and pelvis MRI with/without intravenous contrast agent in colorectal cancer staging: (+) applicable, (−) inapplicable, (−/+) applicable, but highly dependent on scan quality and radiologist experience.

	CT with Intravenous Contrast Agent	MRI (with or without Intravenous Contrast Agent)
X-ray radiation [63]	present	absent
Vulnerability to motion artifacts of small and large intestine [60]	less vulnerable	more vulnerable
Time of the study [60]	shorter	longer
Toleration by the patients [60]	better tolerated	worse tolerated
Soft-tissue contrast resolution (visibility of the rectal wall layers) [60,62,67]	(−)	(+)
Calcifications presence evaluation [60]	(+)	(−)
Image acquisition in the plane perpendicular to the tumour [60]	(−)	(+)
Primary T-stage evaluation (tumour location and size, depth of invasion, EMVI, mesorectal fascia status, anal sphincter involvement) [60,62]	(−)	(+)
Initial local disease assessment (mesorectal and lateral pelvic lymph node involvement) [87]	(−/+)	(+)
Presence of distant metastasis [88]	Lung metastases (+) Liver metastases below 1 cm (−/+) Lymph nodes 15 mm in short axis (+) Peritoneal spread (+)	Lung metastases (−) Liver metastases below 1 cm (+) Lymph nodes 15 mm in short axis (+) Peritoneal spread (+)
Differentiation of mucinous and nonmucinous tumours [62]	(−)	(+)
Differentiation between fibrosis and residual tumour on restating after CTH/RTH [62]	(−)	(+)

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
