# Peer review of "Correlating Ultrastructural Changes in the Invasion Area of Colorectal Cancer with CT and MRI Imaging"

_ijms, 2024, doi:10.3390/ijms25189905_

Round 1

Reviewer 1 Report

Comments and Suggestions for Authors

In the review article entitled “Correlating Ultrastructural Changes in the Invasion Area of Colorectal Cancer with CT and MRI Imaging” the authors summarized the findings about the tumor initiation in the intestine, its invasion into the surrounding areas, extracellular matrix (ECM) disintegration, role of matrix metalloproteinases (MMPs) in the cancer progression, tumor microenvironment (TM) of the colorectal cancer (CRC), the role of tumour-associated macrophages (TAM) and cancer-associated fibroblasts (TAF), and the tumor stiffness and desmoplastic reaction. Subsequently the authors reviewed the role of modern radiological imaging methods, including computer tomography (CT) and magnetic resonance imaging (MRI), and radiomics in the preoperative assessment of the CRC, including the T staging (TNM reporting system) of the invasive CRC tumor. Following are my comments.

1) Minor corrections:

Line 14: Use a better word for “sees”

Line 31: Remove the word “subsequent” if its appropriate.

Line 35: Correct to “healthy columnar epithelium.”, if its appropriate.

Line 36: Correct to “may result in specific structural changes within….”, if its appropriate.

Line 43: Correct to “which is acquired by the transformed cells in this phase….”, if its appropriate.

Line 52: “level of” maybe removed.

Lines 52-53: “to migrate/travel from the epithelium through…”

Line 57: “matrix metalloproteinases”

Lines 57-58: sentence maybe rewritten for better clarity.

Line 59: “are degraded mainly by…”

Line 61: Use the word as either “tumor cells” or “cancer cells”.

Line65: “membrane-type matrix metalloproteinase 14 (MMP-14)”.

Lines 70-71: “matrix metalloproteinases 2”

Lines 71-72: “matrix metalloproteinases 9”

Line 74: “MMP-9 cleaves..”

Lines 74, 82: “increased/elevated expression”

Line 84: “other matrix metalloproteinases including MMP-2, MMP-8,….”

Line 95: improve the words for “a highly composed milieu”

Line 97: “Fragmentation of the epithelial ground substance and its components by MMPs released…”

Line 123: “induces/creates”

Line 128: “periods/stages”

Line 145: What is “TAM processes”?

Line 135-136: “…, and TGF- activity, and generates….”

Lines 150-151: Improve the sentence for better clarity.

Line 175: “…secrete..”

Line 181: “…proteins which subsequently translocate to the nucleus as trimers to bind with the target DNA.”

Lines 181-183: “This transcription factor (SMAD) negatively controls and inhibits cell proliferation and growth via such canonical signaling pathways.”

Line 189: The word “TM cells” looks like a cell type, but in reality, it says “the cells of tumor microenvironment (TM)”. Please correct this appropriately, if needed.

Line 202: “neoplastic tumor”

line 205: provide the full form of “GAG” at its first appearance.

Line 212-213: What is “integrin potential”

Line 241: “….and/or poorly…”

Line 245: “…..is an “expansive pattern”….with…”

Line 248: “…consequently lead to various….”

Lines 245-246: Little more explanation would be better.

Lines 260-262: Improve the sentence for better clarity.

Line 289: “metastases (M)”

Line 296: “..X-ray beams that are attenuated”

Line 325: Use the word as either “tumor cells” or “cancer cells”.

Line 373: “….procedure for rectum cancer…”

Line 395: “…, since the rectum wall…”

Line 414: Use a better word for “overestimating” in this context.

Lines 431-432: Improve the sentence for better clarity.

Line 445: “…ADC values and higher restriction of diffusion when compared….”

Lines 471-472: Improve the sentence for better clarity.

Line 489: “…are employed…”

Line 498: “…present review…”

2) Lines 30-48: References should be given appropriately.

Lines 50-54: Reference maybe provided.

3) Consistency needs to be maintained for the names throughout the manuscript.

3a) The words “large intestine, intestine, colon, bowel” are used randomly. Try to use the words uniformly and appropriately.

3b) The words “matrix metalloproteinases and metalloproteinases” are used randomly.

4) Remove the extra spaces between words in the text: Lines 67, 76, 84, 261, 385, 455.

5) Use the abbreviations and full forms appropriately: The abbreviation along with the full form maybe introduced at its first appearance, and only the abbreviation maybe used later on.

Lines 44, 62, 94, 98, 112, 113: “ECM” for extracellular matrix.

Lines 57, 59, 60, 61, 64, 105: “MMP” for matrix metalloproteinase.

Line 25: The abbreviations “CT” and “MRI” maybe introduced/used here.

Lines 115, 133, 166, 179: “TM” for tumor microenvironment

Line 116: “TAM” tumour-associated macrophages

Line 116: “CAF” for cancer-associated fibroblasts

Lines 135, 146: “VEGF” for vascular endothelial growth factor

6) The tense should be maintained uniformly throughout the manuscript. The present and past tense should not be used randomly.

Lines 74, 76: was

Line 77: is

Line 201: is

Line 205: were

7) Minor edits in the section titles for better suitability:

Line 193: “Neoplasm tumor organization and stiffness”

Line 295: “Computed tomography (CT) in CRC”

Line 355: “Magnetic resonance imaging (MRI) and CRC”

Line 456: “Radiomics and CRC”

8) A table maybe prepared for MMP types, their secretion modes, spectrum of collagen degradation (the target collagen type), cancer stage of expression.

9) A figure may be prepared for the pictorial representation of the healthy/normal tissue microenvironment and the tumor microenvironment of the intestinal wall [mucosa (epithelium, lamina propria, muscularis mucosa), submucosa, muscularis propria], including the component resident (connective tissue originated) and transient (immune system originated) cell populations [mainly the cancer cells, tumour-associated macrophages (TAM), and cancer-associated fibroblasts (CAF)], collagen molecules/fibres, signaling molecules, their activities, and final effects.

10) Tables and legends:

10a) Table 1 title and legend may be improved and better meaningful.

10b) Table 2 title and legend may be improved and better meaningful.

11) The role and applications of computer tomography (CT) and magnetic resonance imaging (MRI), and radiomics in the preoperative and postoperative assessments of the CRC maybe shown in a table or figure. It may include the T staging (T1, T2, T3 and its substages, and T4) of CRC tumor invasion.

12) A diagram may be prepared for the histology of the healthy and cancerous colon and rectal walls, and cancer invasion. Mainly, the tumor histology in colon & rectal walls.

13) A diagram maybe prepared for the CT and MRI images, and radiomics of the T1/T2/T3/T4 stage CRC tumors in colon & rectal cancer.

14) Questions: Do the corrections appropriately in the manuscript.

Line 311: is it “96%, 89%, 79%, and 75%, respectively” ?

Line 428: what about 10-15mm depth of the invasion, and which T-staging (T3 substage) does it belong.

Line 454: “The abdomen and pelvis CT” is not discussed anywhere in the main text.

Line 494: Is it “preoperative” ?

Author Response

Dear Professor,

I am grateful for your time and detailed review, it was very helpful. The suggestions you made are important and insightful. Me and co-authors have discussed them and did our best to address it as precisely as possible to improve the article. I have learn a lot while reading your report and I know that your advice will help me with writing review and research studies in the future.

We have prepared three diagrams to better illustrate the two patterns of cancer invasion, processes leading to cancer promotion and the MRI evaluation of T-stages of CRC.

I hope that after following your suggestions the review article will present the topic more accurately and become more comprehensive. I write the comments on your report below.

Kind regards,

Joanna Urbaniec-Stompór

1) Minor corrections:

Line 14: Use a better word for sees”.

Line 14: The word „sees” was changed to „causes”.

Line 31: Remove the word subsequent” if its appropriate.

Line 31: The word subsequent” was removed.

Line 35: Correct to healthy columnar epithelium.”, if its appropriate.

Line 35: Correction to healthy columnar epithelium” was made.

Line 36: Correct to may result in specific structural changes within….”, if its appropriate.

Line 36: Correction to may result in specific structural changes within” was made.

Line 43: Correct to which is acquired by the transformed cells in this phase….”, if its appropriate.

Line 43: Correction to which is acquired by the transformed cells in this phase….” was made.

Line 52: level of” maybe removed.

Line 52: The words level of” was removed.

Lines 52-53: to migrate/travel from the epithelium through…”

Lines 52-53: The word „pass” was changed tomigrate”.

Line 57: matrix metalloproteinases”

Line 57: Correction to plural matrix metalloproteinases” was made.

Lines 57-58: sentence maybe rewritten for better clarity.

Lines 57-58: The sentence was changed to: „Matrix metalloproteinases (23 enzymes in human tissue) degrade different types of collagen structures.’’

Line 59: are degraded mainly by…”

Line 59: Word „via” was corrected to by”.

Line 61: Use the word as either tumor cells” or cancer cells”.

Line 61: „Tumour cancer cells” was corrected tocancer cells”.

Line 65: membrane-type matrix metalloproteinase 14 (MMP-14)”.

Line 65: The correction to membrane-type matrix metalloproteinase 14 (MMP-14)” was made.

Lines 70-71: matrix metalloproteinases 2”

Lines 71-72: matrix metalloproteinases 9”

Lines 70-71: The correction to matrix metalloproteinases 2” was made.

Lines 71-72: The correction to matrix metalloproteinases 9” was made.

Line 74: MMP-9 cleaves..”

Line 74: The word „cleave” was changed tocleaves”.

Lines 74, 82: increased/elevated expression”

Lines 74, 82: The word „upper” was changed to increased”.

Line 84: other matrix metalloproteinases including MMP-2, MMP-8,….”

Line 84: The correction to other matrix metalloproteinases including MMP-2, MMP-8” was made.

Line 95: improve the words for a highly composed milieu”

Line 95: Words a highly composed milieu” were changed to a highly composed environment”.

Line 97: Fragmentation of the epithelial ground substance and its components by MMPs released…”

Line 97: The correction to Fragmentation of the epithelial ground substance and its components by MMPs released…” was made.

Line 123: induces/creates”

Line 123: The word induces” was corrected to „creates”.

Line 128: periods/stages”

Line 128: The word periods” was corrected to „stages”.

Line 145: What is TAM processes”?

Line 145:  The sentence was changed to: „ Activated M2 macrophages produce numerous chemokines and growth factors that induce numerous tumor-promoting processes.”.

Line 135-136: “…, and TGF- activity, and generates….”

Line 135-136: The correction to “…, and TGF- activity, and generates….” was made.

Lines 150-151: Improve the sentence for better clarity.

Lines 150-151: The sentence from lines 150-151 was changed to: „Generally, if M2 macrophages are activated in tumour microenvironment, they become involved in cancer cell invasion promotion, despite their physiological pro-healing role.”

Line 175: “…secrete..”

Line 175: The word was corrected to secrete”.

Line 181: “…proteins which subsequently translocate to the nucleus as trimers to bind with the target DNA.”

Line 181: The correction was made.

Lines 181-183: This transcription factor (SMAD) negatively controls and inhibits cell proliferation and growth via such canonical signaling pathways.”

Lines 181-183: The correction was made.

Line 189: The word TM cells” looks like a cell type, but in reality, it says the cells of tumor microenvironment (TM)”. Please correct this appropriately, if needed.

Line 189: The sentence was changed to: „Furthermore, it should be noted that TGF-β is secreted by the cells of tumor microenvironment: (cancer cells, TAMs and CAFs) with autocrine and paracrine signalling, and finally, such multifactorial positive feedback promotes invasive CRC phenotypes at the initial and advanced stages of illness”.

Line 202: neoplastic tumor”

Line 202: The words „neoplasm tumor” was changed to neoplastic tumor”.

line 205: provide the full form of GAG” at its first appearance.

line 205: The full form of GAG” was provided.

Line 212-213: What is integrin potential”

Line 212-213: The sentence was changed to „Cancer cells interact with stroma components by the integrins (adhesion glycoproteins).”.

Line 241: “….and/or poorly…”

Line 241: The correction to “….and/or poorly…” was made.

Line 245: “…..is an expansive pattern”….with…”

Line 245: The correction to “…..is an expansive pattern”….with…” was made.

Line 248: “…consequently lead to various….”

Line 248: The correction to “…consequently lead to various….” was made.

Lines 245-246: Little more explanation would be better.

Lines 245-246: Additional information was added: „The boundary between the tumour mass and normal tissue is formed by collagen fibers aggregation.”

Lines 260-262: Improve the sentence for better clarity.

Lines 260-262: The sentence was changed to „ The presence of myxoid stroma (amorphous mucinous substance), keloid-like collagen (thick collagen bundle eosinophilic hyalinization) and fine, multilayer collagen fibres, was described as an immature, intermediate or mature desmoplastic reaction respectively.”.

Line 289: metastases (M)”

Line 289: The correction to metastases (M)” was made.

Line 296: X-ray beams that are attenuated”

Line 296: The correction to X-ray beams that are attenuated” was made.

Line 325: Use the word as either tumor cells” or cancer cells”.

Line 325: The word tumor cells” was used.

Line 373: “….procedure for rectum cancer…”

Line 373: The correction to “….procedure for rectum cancer…” was made.

Line 395: “…, since the rectum wall…”

Line 395: The correction to “…, since the rectum wall…” was made.

Line 414: Use a better word for overestimating” in this context.

Line 414: A word overestimating” was changed to „overstaging”.

Lines 431-432: Improve the sentence for better clarity.

Lines 431-432: The sentence was change to: These neoplasm tumours demonstrate variable enhancement after gadolinium-based intravenous contrast administration and may show restricted diffusion on diffusion-weighted imaging (DWI).

Line 445: “…ADC values and higher restriction of diffusion when compared….”

Line 445: The correction to “…ADC values and higher restriction of diffusion when compared….” was made.

Lines 471-472: Improve the sentence for better clarity.

Lines 471-472: The sentence was corrected to: „T-staging of a rectal cancer using radiomics was more accurate than staging performed by two experienced radiologist using MRI.”

Line 489: “…are employed…”

Line 489: The correction to are employed” was made.

Line 498: “…present review…”

Line 498: The correction to present review” was made.

2) Lines 30-48: References should be given appropriately. Lines 50-54: Reference maybe provided.

2: Additional citation was added to paragraph 1.

3) Consistency needs to be maintained for the names throughout the manuscript.

3a) The words large intestine, intestine, colon, bowel” are used randomly. Try to use the words uniformly and appropriately.

3b) The words matrix metalloproteinases” and „metalloproteinases” are used randomly.

3, 3a, 3b: The word „large intestine” was applied uniformly. The word „matrix metalloproteinases” was applied uniformly.

4) Remove the extra spaces between words in the text: Lines 67, 76, 84, 261, 385, 455.

4: The extra spaces have been removed.

5) Use the abbreviations and full forms appropriately: The abbreviation along with the full form maybe introduced at its first appearance, and only the abbreviation maybe used later on.

Lines 44, 62, 94, 98, 112, 113: ECM” for extracellular matrix.

Lines 57, 59, 60, 61, 64, 105: MMP” for matrix metalloproteinase.

Line 25: The abbreviations CT” and MRI” maybe introduced/used here.

Lines 115, 133, 166, 179: TM” for tumor microenvironment

Line 116: TAM” tumour-associated macrophages

Line 116: CAF” for cancer-associated fibroblasts

Lines 135, 146: VEGF” for vascular endothelial growth factor

5: Abbreviations have been used after being introduced at its first appearance.

6) The tense should be maintained uniformly throughout the manuscript. The present and past tense should not be used randomly.

Lines 74, 76: was

Line 77: is

Line 201: is

Line 205: were

6: The tense mistakes was corrected.

7) Minor edits in the section titles for better suitability:

Line 193: Neoplasm tumor organization and stiffness”

Line 295: Computed tomography (CT) in CRC”

Line 355: Magnetic resonance imaging (MRI) and CRC”

Line 456: Radiomics and CRC”

7: Thank you very much for the edits, the section titles become more accurate.

8) A table maybe prepared for MMP types, their secretion modes, spectrum of collagen degradation (the target collagen type), cancer stage of expression.

8: As tables focusing on MMPs are part of many reference articles and are easy to find by a reader we decided to describe it in a text paragraph. We would like to draw readers’ attention to importance of different biomarkers potentially used to predict patients’ survival and comparison of CT and MRI in CRC evaluation.

9) A figure may be prepared for the pictorial representation of the healthy/normal tissue microenvironment and the tumor microenvironment of the intestinal wall [mucosa (epithelium, lamina propria, muscularis mucosa), submucosa, muscularis propria], including the component resident (connective tissue originated) and transient (immune system originated) cell populations [mainly the cancer cells, tumour-associated macrophages (TAM), and cancer-associated fibroblasts (CAF)], collagen molecules/fibres, signaling molecules, their activities, and final effects.

9: Thank you for the suggestion. We have prepared a figure to better illustrate the role of TAM, CAF and their signaling molecules in a cancer proliferation process.

10) Tables and legends:

10a) Table 1 title and legend may be improved and better meaningful.

10b) Table 2 title and legend may be improved and better meaningful.

10, 10a, 10b: Table 1 title have been changed to: „The potential molecular and cellular biomarkers and histological features observed in the pathomorphological assessment. Chosen features are related with colorectal cancer invasion, finally they are predictors of patients’ shorter survival.” Table 2 title and legend have been changed to:Analysis of potential applications and limitations of chest, abdomen and pelvis CT with intravenous contrast agent and abdomen and pelvis MRI with/without intravenous contrast agent in colorectal cancer staging. (+) - applicable (-) - inapplicable, (-/+) - applicable, but highly dependent on scan quality and radiologist experience.”

11) The role and applications of computer tomography (CT) and magnetic resonance imaging (MRI), and radiomics in the preoperative and postoperative assessments of the CRC maybe shown in a table or figure. It may include the T staging (T1, T2, T3 and its substages, and T4) of CRC tumor invasion.

11: The applications and limitations of CT and MRI have been discussed in Table 2. As radiomics have not yet been introduced in clinical practice it seems to be more accurate to discuss it as a future tool.

An additional paragraph in „Radiomics and CRC” section was added, about blood oxygenation level dependent (BOLD) MRI and tumour oxygenation level dependent (TOLD) MRI.

We have prepared a diagram to illustrate the anatomy of the rectum and the possible locations of rectal cancer, along with corresponding T categories. 

12) A diagram may be prepared for the histology of the healthy and cancerous colon and rectal walls, and cancer invasion. Mainly, the tumor histology in colon & rectal walls.

12: Thank you for the suggestion. The diagram illustrating histology of the healthy large intestine wall and two growth patterns (expansive, infiltrative) of cancer invasion have been prepared.

13) A diagram maybe prepared for the CT and MRI images, and radiomics of the T1/T2/T3/T4 stage CRC tumors in colon & rectal cancer.

13: Thank you for the suggestion. The diagram to illustrate the anatomy of the rectum and the possible locations of rectal cancer, along with corresponding T categories have been prepared. 

14) Questions: Do the corrections appropriately in the manuscript.

Line 311: is it 96%, 89%, 79%, and 75%, respectively” ?

The double-check in reference article have been made and the cited data are correct: „96%, 89%, 75%, and 79%, respectively”. It may be due to difficulties with T3 substages assessment at the moment of diagnosis and more aggressive treatment of T4 tumours.

Line 428: what about 10-15mm depth of the invasion, and which T-staging (T3 substage) does it belong.

Line 428: The sentence was rewritten with the correct T3-substages: T3a <1mm, T3b 1-5mm, T3c 5-15mm, T3d>15mm.

Line 454: The abdomen and pelvis CT” is not discussed anywhere in the main text.

Line 454: It was clarified that whole paragraph discussing the CT applies to the chest, abdomen and pelvis contrast-enhanced CT in a rewritten sentence in a line 303: „Abdomen and pelvis contrast-enhanced CT is a standard modality used for the CRC preoperation characteristics: T- (localisation and size of tumour), N- and M- staging evaluation.”

Line 494: Is it preoperative” ?

Line 494: Yes, it was clarified by the change in the sentence: The preoperative staging of local cancer advancement (the T parameter of the TNM staging system) and type of cancer invasion into the large intestine wall allow for the determination of the neoplasm's development and the prediction of the course of illness.

Reviewer 2 Report

Comments and Suggestions for Authors

International Journal of Molecular Sciences (Manuscript ID: ijms-3145423), Comments to the Authors:

Title: Correlating Ultrastructural Changes in the Invasion Area of Colorectal Cancer with CT and MRI Imaging

Comments

The submitted review highlighted on the processes occurring at the “front of cancer invasion”, with a particular focus on the role of the desmoplastic reaction in neoplasm development. It correlated the findings from the microscopic observation of the cancer's ultrastructure with the potential of modern radio-logical imaging (such as computer tomography and magnetic resonance imaging), which visualizes the tumor, its boundaries, and the tissue reactions in the large intestine.

I think the submitted review can be accepted after the authors respond to the following comments: 

1.     The authors should indicate the rationale behind writing the review. The authors should explain why they initiated the idea of the review.

2.     The authors should provide a more in-depth analysis of their findings. They should highlight the most interesting results reported in certain papers in comparison to other results in other reports to give the reader a better understanding of the actual activity.

3.     The authors can draw some figures summarizing the individual cancer development processes.

4.     The authors should provide their insights on the potential future applications of the CT and MRI imaging in the detection of the invasion area of colorectal cancer.

5.     The authors should highlight their contribution to the research on CT and MRI imaging.

6.     The authors should indicate their approach in collecting literature, what were the search engines used in collecting literature, what were the keywords used in collecting literature, and what was the time span covered by the review.

7.     There are some typos and grammatical errors that should be corrected.

Author Response

Dear Professor,

I am grateful for your time and insightful suggestions. Together with co-authors we have discussed it and did our best to address them as precisely as possible to improve the article. We have prepared three diagrams to better illustrate the two patterns of cancer invasion, processes leading to cancer promotion and the MRI evaluation of T-stages of CRC.

I hope that after following your suggestions the review article will present the topic more accurately and become more comprehensive. I write a comment on your report below. Thank you very much once again.

Kind regards,

Joanna Urbaniec-Stompór

1.     The authors should indicate the rationale behind writing the review. The authors should explain why they initiated the idea of the review.

1: An additional explanation have been added to first paragraph: „As process of CRC invasion in patientspopulation varies, the aim of modern personalized medicine is to identify the patients with more aggressive course of the disease to administer appropriately intensive treatment. The aim of this review is to present the process of CRC promotion and progression at ultrastructural level and its relation to modern diagnostic imaging (CT, MRI) routinely performed at the initial stage of the diagnosis. The presentation of histological features and its reference to the macroscopic image may be helpful at the initial diagnosis with the selection of appropriate treatment.”

2.   The authors should provide a more in-depth analysis of their findings. They should highlight the most interesting results reported in certain papers in comparison to other results in other reports to give the reader a better understanding of the actual activity.

2: The aim of this study is to summarize current knowledge in order to to plan future research study that will attempt to correlate these two different levels of observation and knowledge in relation to cancer growth (ultrastructural vs. macroscopic diagnostic imaging). We hope that the knowledge acquired on ultrastructure of cancer promotion and progression will cause the development of imaging techniques in the future. In future research the direction of these studies will be developed.

3.  The authors can draw some figures summarizing the individual cancer development processes.

3: Three diagrams have been prepared: first focuses on histology of the healthy large intestine wall and the two growth patterns (expansive, infiltrative) of cancer invasion. Second applies to the role of TAM, CAF and their signaling molecules in a cancer proliferation process. Third diagram is made to to illustrate the anatomy of the rectum and the possible locations of rectal cancer, along with corresponding T categories.

4.     The authors should provide their insights on the potential future applications of the CT and MRI imaging in the detection of the invasion area of colorectal cancer.

4: Additional information concerning future of MRI imaging and radiomics have been added to paragraph „Radiomics and CRC”: „In the future, MRI applications could include blood oxygenation level dependent (BOLD) MRI and tumour oxygenation level dependent (TOLD) MRI, which could have a potential to identify, spatially map and quantify tumour hypoxia, which occurs as an effect of a disordered angiogenesis. After improving standardization, validation, and reproducibility, and with prospective multicentric study trials, CT and MRI radiomics may become introduced to the clinical routine, to help staging patients who require neoadjuvant chemotherapy”.

5.    The authors should highlight their contribution to the research on CT and MRI imaging.

5: Additional information concerning research on CT and MRI imaging have been highlighted in the Authors Contributions section.

6.     The authors should indicate their approach in collecting literature, what were the search engines used in collecting literature, what were the keywords used in collecting literature, and what was the time span covered by the review.

6.  The literature has been collected between April 2024 July 2024. Our database included Pubmed and the keywords used at the initial reaserch were: desmoplastic reaction, cancer-associated fibroblasts, colorectal cancer, ECM, tumour-associated macrophages, tumor microenvironment, cancer biomarkers, computer tomography, radiomics in colorectal cancer, TNM cancer staging, layers of rectal wall in CT and MRI. The keywords used at additional research: LASSO analysis, front of invasion, ESGAR guidelines, CT colonography, MMP enzymes, M2 macrophages.

7.     There are some typos and grammatical errors that should be corrected.

7.  The double-check of spelling, vocabulary and grammatical mistakes have been made and corrected.

Reviewer 3 Report

Comments and Suggestions for Authors

I suggest accepting the work in its current form.

Author Response

Dear Professor,

Thank you very much for your suggestion. I am very grateful for your time and positive feedback. 

Kind Regards,

Joanna Urbaniec-Stompór

Round 2

Reviewer 1 Report

Comments and Suggestions for Authors

The authors addressed all the comments well, made the appropriate corrections in the text, and prepared few figures, as suggested, to better illustrate some contents of the manuscript. The manuscript is improved a lot. However, the following corrections should be taken care.

Major comments:

1) The figures are not cited anywhere in the manuscript text. So, please cite all the three figures and two tables in the manuscript text appropriately.

2) Figures 2 and 3: Label the different layers of the wall of large intestine and rectum in a better way in the figures 2 and 3, respectively.

Minor comments:

Line 45: Remove “which is”.

Line 95: Remove the punctuation mark “:”

Line 96: Replace “was” with “are”.

Lines 148-149: Correct “TGF- activity” to “TGF-β activity”.

Lines 196-198: Remove the extra space.

Line 205: Remove the punctuation mark “:”

Line 229: Replace “by” with “through/via”.

Lines 259-262: Remove the extra space.

Lines 289-290: Correct to “were described as immature, intermediate, and mature desmoplastic reactions, respectively”

Line 511: “radiologists”

Lines 50-57 and 59-63: Add references, if needed.

Lines 147-148 and 159-160: The abbreviation along with the full form maybe introduced at its first appearance.

Lines 85-87, 217, and 221: The tense should be maintained uniformly throughout the manuscript. The present and past tense should not be used randomly. [Previous Comment]

Lines 459-460: Correct to “T3a < 1mm, T3b = 1-5mm, T3c = 5-15mm, and T3d > 15mm [67,82].”

Line 461: Figure 3 legend: “Illustration of different T-stages of CRC and their relation to layers of rectum.”

Lines 488-491: Figure 2 legend: Use the punctuation marks appropriately to make it better meaningful.

Author Response

Dear Professor,

Thank you once again for the time you have spent on the work with the review, you had a great impact on improving both its content and arrangement. Me and my colleagues were doing our best to address the valuable comments you made.

We corrected the figures to be better meaningful. I hope that minor tense and punctuation mistakes are now corrected.

It is a honor to have an opportunity to cooperate in international experts team.

Kind Regards,

Joanna Urbaniec-Stompór

  1. The figures are not cited anywhere in the manuscript text. So, please cite all the three figures and two tables in the manuscript text appropriately.

The citation of three figures and two tables have been improved.

2) Figures 2 and 3: Label the different layers of the wall of large intestine and rectum in a better way in the figures 2 and 3, respectively.

Additional arrows have been added to clarify the figures.

Line 45: Remove which is”.

It has been removed.

Line 95: Remove the punctuation mark :”

It has been removed.

Line 96: Replace was” with are”.

It has been replaced.

Lines 148-149: Correct TGF- activity” to TGF-β activity”.

The correction was made.

Lines 196-198: Remove the extra space.

The extra space was removed.

Line 205: Remove the punctuation mark :”

The punctuation mark was removed.

Line 229: Replace by” with through/via”.

It was replaced.

Lines 259-262: Remove the extra space.

The extra space was removed.

Lines 289-290: Correct to were described as immature, intermediate, and mature desmoplastic reactions, respectively”

The correction was made.

Line 511: radiologists”

The correction was made.

Lines 50-57 and 59-63: Add references, if needed.

The paragraph aims to express the intention of the review and refers to general knowledge emphasized in various cited reference articles, so after discussion we think that it doesnt require additional citation.

Lines 147-148 and 159-160: The abbreviation along with the full form maybe introduced at its first appearance.

The full form has been added to first appearance of „VEGF”.

Lines 85-87, 217, and 221: The tense should be maintained uniformly throughout the manuscript. The present and past tense should not be used randomly. [Previous Comment]

The tense have been corrected to present simple.

Lines 459-460: Correct to T3a < 1mm, T3b = 1-5mm, T3c = 5-15mm, and T3d > 15mm [67,82].”

The correction was made.

Line 461: Figure 3 legend: Illustration of different T-stages of CRC and their relation to layers of rectum.”

The correction was made.

Lines 488-491: Figure 2 legend: Use the punctuation marks appropriately to make it better meaningful.

The correction of punctuation marks was made to:  Analysis of potential applications and limitations of chest, abdomen and pelvis CT with intravenous contrast agent and abdomen and pelvis MRI with/without intravenous contrast agent in colorectal cancer staging: (+) applicable, (-) inapplicable, (-/+) applicable, but highly dependent on scan quality and radiologist experience.”

Reviewer 2 Report

Comments and Suggestions for Authors

International Journal of Molecular Sciences (Manuscript ID: ijms-3145423 - Revised Review), Comments to the Authors:

Title: Correlating Ultrastructural Changes in the Invasion Area of Colorectal Cancer with CT and MRI Imaging

Comments

After reading the authors response to me comments, I think the revised review can be published.

Author Response

Dear Professor,

Thank you very much once again for your valuable comments and suggestions. We were doing our best to improve the review article and feel very grateful for your acceptance.

Yours faithfully,

Joanna Urbaniec-Stompór